# Effect of Doping on the Bandgap of the Organic–Inorganic Hybrid Ferroelectric Material [C$_6$N$_2$H$_{18}$]Bi$_{1−x}$Sb$_x$I$_5$ (0.0 < x < 1.0)

Xin Guo [1,2], Jialin Zhu [1,3,*], Xiaoping Zou [1,4], Wenqi Huang [2], Chunqian Zhang [1], Zixiao Zhou [1], Junqi Wang [1], Hao Wang [1] and Hanmiao Zhang [1]

1   Research Center for Sensor Technology, Beijing Key Laboratory for Sensor, Jianxiangqiao Campus, Beijing Information Science and Technology University, Beijing 100101, China
2   School of Science, Jianxiangqiao Campus, Beijing Information Science and Technology University, Beijing 100101, China
3   School of Automation, Jianxiangqiao Campus, Beijing Information Science and Technology University, Beijing 100101, China
4   Beijing Advanced Innovation Center for Materials Genome Engineering, Jianxiangqiao Campus, Beijing Information Science and Technology University, Beijing 100101, China
*   Correspondence: jlzhu@bistu.edu.cn; Tel.: +86-185-1476-6686

**Abstract:** The rapidly developing organic–inorganic hybrid chalcogenide solar cells have now become a hot topic of interest. However, the bandgap of inorganic ferroelectric materials with a typical chalcogenide structure is too wide to match the solar spectrum, while the ferroelectricity of organic-inorganic hybrid chalcogenide materials with a narrow bandgap, meth amide–lead–iodine, is not obvious, and the lead element causes environmental pollution. The recently discovered organic–inorganic hybrid material [C$_6$N$_2$H$_{18}$]BiI$_5$ with good ferroelectricity and the narrowest bandgap of molecular ferroelectrics can absorb visible light in the range of 380 nm to 660 nm, and compound [C$_6$N$_2$H$_{18}$]SbI$_5$ with the Bi cognate element Sb was also synthesized. In this paper, we designed the first experiment to prepare thin films by mixing and doping the above two materials in five different molar ratios, and we comparatively studied the changes in crystal structure, surface morphology, and photophysical properties of the prepared multicomponent hybrid films according to the mixing ratio. A theoretical model was developed to calculate and analyze the bandgap of the hybrid doped compounds and compare it with the experimental values. It was found that the absorption spectra of the multicomponent hybrid films were red-shifted relative to the original material, indicating that the forbidden bandwidth was reduced to absorb a wider range of visible light, and the reason for this was thought to be the narrowing of the bandgap due to doping. When the mixing ratio was 0.4:0.6, the bandgap was the narrowest and the light absorption was the best; the highest quality of the film was obtained when the mixing ratio was 0.5:0.5.

**Keywords:** perovskite solar cell; multicomponent doping; forbidden bandwidth

## 1. Introduction

With the decrease in fossil fuels, green and renewable solar energy has become an important part of the energy for human use. In recent years, organic–inorganic hybrid chalcogenide solar cells have become a hot research topic [1–4]. However, the high cost, low stability, and environmental pollution due to their lead content are non-negligible problems. Typical inorganic oxide chalcogenide ferroelectric materials have the problems of too wide a bandgap to match the solar spectrum and a more complicated preparation process [5]. Therefore, in this paper, a novel organic–inorganic hybrid molecular ferroelectric material [C$_6$N$_2$H$_{18}$]BiI$_5$ (HDA-BiI$_5$) was chosen, which not only has the narrowest bandgap of 1.89 eV of current molecular ferroelectric materials and can absorb visible light in the range of 380–660 nm, but also has good ferroelectricity, representing a nonpolluting, low-cost, and stable ferroelectric photovoltaic material [6].

A high-quality ferroelectric film is a prerequisite for the preparation of high-performance ferroelectric solar cells, which requires a large grain size, good crystallinity, and a flat and dense film structure [7]. Currently, the recognized bandgap of the light-absorbing layer of photovoltaic devices is in the range of 1.4–1.5 eV [8], and regulating the forbidden bandwidth of ferroelectric materials and preserving their good ferroelectricity are the focus of research. Elemental substitution and material mixing have been favored by researchers as frequently used means in materials research, usually to change the forbidden bandwidth of materials or to enhance the quality of thin films [9–12]. Ying Luo investigated the modulation of the electronic structure of the two-dimensional ferroelectric semiconductor $\alpha$-In$_2$Se$_3$ by alternative doping of a single main group element using a first-principles calculation system based on density generalized theory. It was proposed that the nonvolatile regulation of the electronic structure of the system by the external electric field is achieved by the asymmetric doping of two In atomic layers with different coordination environments present in $\alpha$-In$_2$Se$_3$ and the interchangeable properties of the coordination environments of the two In atomic layers corresponding to their ferroelectric flipping [13]. Qing Zhang prepared La-doped Bi$_4$Ti$_3$O$_{12}$ ferroelectric thin films using a sol–gel method to study the effects of different La doping amounts on the microstructure and photoelectric properties of Bi$_4$Ti$_3$O$_{12}$ ferroelectric thin films. The results showed that, with the increase in La element doping, the diffraction peak gradually shifted to a higher angle, the grain size gradually decreased, the absorption edge was blue-shifted, and the forbidden bandwidth slightly increased. The photoelectric response of the La-doped Bi$_4$Ti$_3$O$_{12}$ films was significantly better than that of Bi$_4$Ti$_3$O$_{12}$ [14]. The Fe-doped barium titanate BaTi$_{1-x}$Fe$_x$O$_3$ ($0.10 \le x \le 0.60$) samples were prepared using a high-temperature solid-phase reaction method, and their structural, ferroelectric, and magnetic properties were investigated by Sun Huilai et al. The BaTi$_{1-x}$Fe$_x$O$_3$ material has both ferroelectric and ferromagnetic properties at room temperature. With the increase in Fe doping from 10% to 60%, the magnetic moment generated by a single Fe ion in the material increased from 0.70 μB to 1.55 μB [15].

In this paper, the organic–inorganic hybrid material [C$_6$N$_2$H$_{18}$]SbI$_5$ [16] was synthesized by replacing the bismuth element in the ferroelectric material with its cognate antimony element, and the two ferroelectric materials were mixed in different ratios so that the ratios of Bi and Sb elements in different samples showed gradient changes; the structure and morphology of the multicomponent ferroelectric hybrid films were studied comparatively, the photophysical properties of the prepared ferroelectric films were characterized, including absorption spectra and steady-state fluorescence spectra, and a theoretical model was developed to calculate the bandgap of the mixed doped compounds for comparison with the experimental values to study the effect of multicomponent mixing on the bandgap.

## 2. Experiment

*Preparation of Multicomponent Hybrid Ferroelectric Films*

The preparation of [C$_6$N$_2$H$_{18}$]BiI$_5$ and [C$_6$N$_2$H$_{18}$]SbI$_5$ ferroelectric powders is shown in the supporting literature.

(1) The prepared [C$_6$N$_2$H$_{18}$]BiI$_5$ and [C$_6$N$_2$H$_{18}$]SbI$_5$ powders were removed and placed on a hot plate for heating. The purpose of this step was to remove the moisture present in the powders due to moisture and the excess HI.

(2) The molecular masses of the two materials [C$_6$N$_2$H$_{18}$]BiI$_5$ and [C$_6$N$_2$H$_{18}$]SbI$_5$ were calculated and mixed in different molar ratios of 0.8:0.2, 0.6:0.4, 0.5:0.5, 0.4:0.6, and 0.2:0.8.

(3) The mixed powder was completely dissolved in DMF solution at a concentration of 500 mg/mL. Magnetic stirring or sonication could be used to speed up this process.

(4) The completely dissolved ferroelectric mixture was filtered through a 0.22 μm filter to obtain a solution for spin-coating.

(5) The spin-coating method is described in the supporting literature and is not repeated here.

## 3. Results and Discussion

### 3.1. X-ray Diffraction Characterization of Single Ferroelectric Films

The X-ray diffraction spectra of two ferroelectric materials, HDA-BiI$_5$ and [C$_6$N$_2$H$_{18}$]SbI$_5$, formed as thin films on glass/ITO substrates are given here for a structural analysis and comparison with the literature. According to reports in the literature [16,17], [C$_6$N$_2$H$_{18}$]BiI$_5$ has a bandgap of 1.89 eV, which is claimed to be the narrowest bandgap among the known ferroelectrics, implying an absorption range between 380 nm and 660 nm, capable of absorbing most of the UV and visible light. [C$_6$N$_2$H$_{18}$]SbI$_5$, a molecular ferroelectric electrode with a similar structure to [C$_6$N$_2$H$_{18}$]BiI$_5$, is being explored by researchers at the initial stage [13]. The blue curve in Figure 1 shows the X-ray diffraction spectrum of HDA-BiI$_5$, and the crystallographic indices (110), (111), (120), (320), (322), (330), (113), and (440) of the ferroelectric material can be observed, which almost exactly match the XRD of the HDA-BiI$_5$ film mentioned in the literature, indicating a good reduction of the ferroelectric materials in the literature. The red curve shows the X-ray diffraction spectrum of [C$_6$N$_2$H$_{18}$]SbI$_5$, and its peak position and peak intensity are in general agreement with the X-ray diffraction spectrum (PXRD) of polycrystalline powder in the literature (the XRD spectrum of this material is not given in the literature). Through observation and comparison, it can be found that both [C$_6$N$_2$H$_{18}$]SbI$_5$ and [C$_6$N$_2$H$_{18}$]BiI$_5$ had the highest peak intensities at a crystal orientation index of (110), i.e., the main peaks were in agreement.

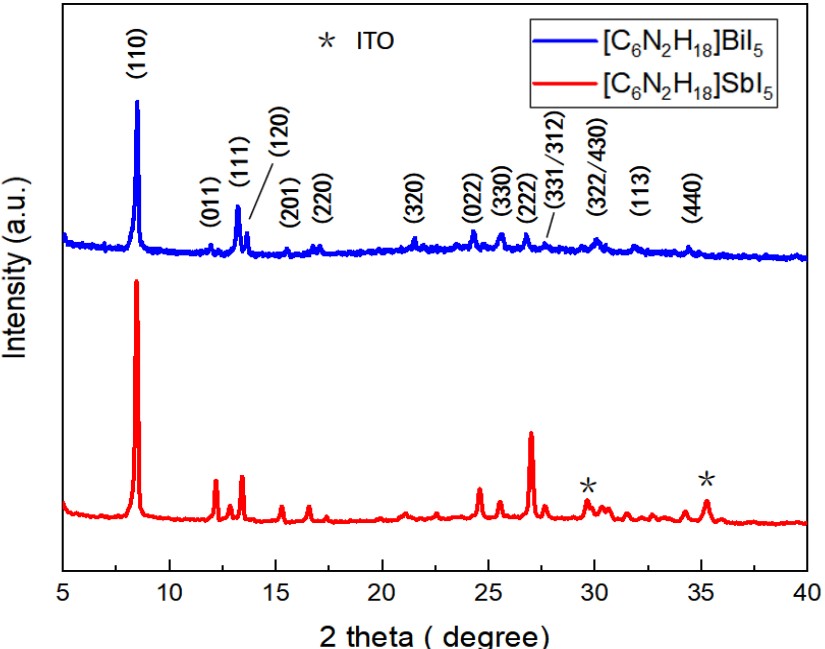

**Figure 1.** X-ray diffraction spectrum of single organic–inorganic hybrid ferroelectric materials. Where * is the diffraction peak of the ITO material.

### 3.2. XRD Characterization of Multicomponent Hybrid Ferroelectric Films

A comparison of the X-ray diffraction patterns of multicomponent ferroelectric hybrid films (deposited on glass/ITO substrates) with different molar ratios is shown in Figure 2. The selected mixing ratios were 0.8:0.2, 0.6:0.4, 0.5:0.5, 0.4:0.6, and 0.2:0.8. Peaks with crystallographic indices of (110), (111), (120), (320), (040), (330), (113), and (440) were found in the X-ray diffraction spectra of each mixing ratio at almost all positions coinciding with the two unmixed original materials. It can be observed that the peak positions and peak intensities of the materials changed significantly at $2\theta = 12.5°$, $2\theta = 26.5°$, $2\theta = 30°$, and $2\theta = 34.5°$ upon increasing the proportion of Sb elements. By comparison, it can be found that the peak intensity of ferroelectric materials containing Sb elements was generally higher than that of the pure [C$_6$N$_2$H$_{18}$]BiI$_5$ material, and the peak intensity of the characteristic peak ($2\theta = 8°$)

with a crystallographic index of (110) was higher than that of the two pure ferroelectric materials when the mixing ratios were 0.6:0.4 and 0.2:0.8. When the mixing ratio was 0.6:0.4, the peak intensities of the X-ray diffraction peaks of the materials were generally higher than those of the unmixed ferroelectric materials.

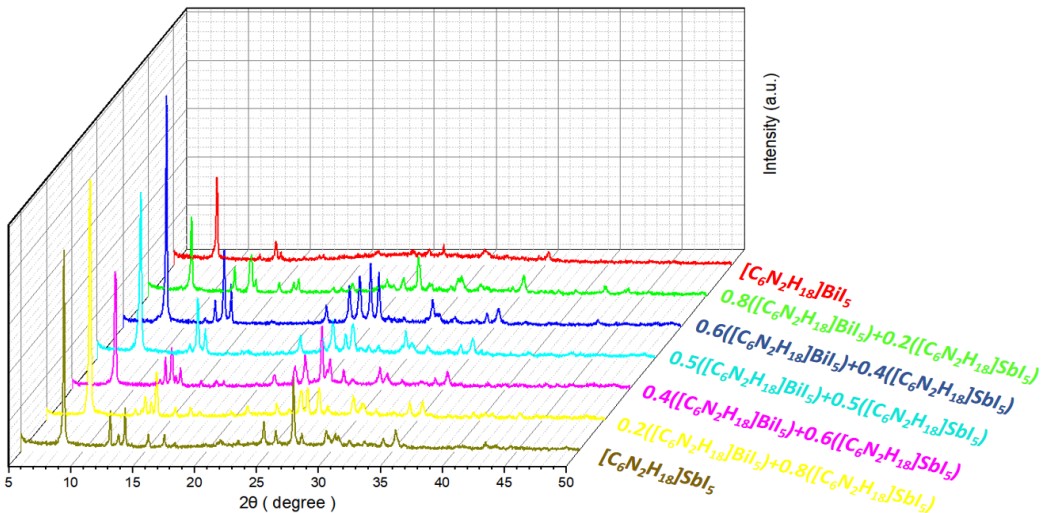

**Figure 2.** X-ray diffraction spectra of ferroelectric materials with different mixing ratios.

The atomic radius of the Sb element is smaller than that of the Bi element, and, in the multicomponent mixed film, the Sb element partially replaces the Bi element. This leads to the distortion of the crystal structure and, therefore, alters the X-ray diffraction peak position and intensity (Figure 2).

The XRD pattern of Figure 2 was imported into Materials Studio software for refinement using the Rietveld method, and the established theoretical cell structure was fine-tuned. The actual crystal structure parameters, including lattice constants and atomic positions, were obtained (please refer to the attached crystal structure parameters).

### 3.3. Study on the Morphology of Multicomponent Hybrid Ferroelectric Films

The top SEM views of the above five ferroelectric films are shown in Figure 3, while the left column shows the morphology of these five films at 15,000× magnification, and the right column shows the morphology of these five films at 50,000× magnification. From Figure 3a, it can be seen that, when the mixing ratio of Sb and Bi elements was 0.8:0.2, a large number of cracks appeared on the surface of the ferroelectric films with a maximum width of several tens of nanometers, and the whole films showed a cracked state; from the morphological view at high magnification, it can be observed that there were still pores of different depths on the surface of the films at this time, and their diameters were less than 100 nm. In the $C_6N_2H_{18}]SbI_5$ film (Figure S1c), a large number of cracks appeared, the number and diameter of pinholes decreased, and the flatness of the film was slightly improved. With the increase in the molecular percentage of Bi elements, it can be observed from Figure 3c,e,g that the surface of the films had several cracks with a width of several tens of nanometers, without the presence of pores; by observing the morphological picture at high magnification, it can be found that the ferroelectric films had a dense and flat surface, and their film quality was improved, although cracks still existed. When the mixing ratio was 0.2:0.8, i.e., the highest content of Bi, the film surface was almost crack-free, but some accumulation of grains and some pinholes with diameters ranging from 20 nm to 50 nm appeared, which reduced the flatness of the film surface.

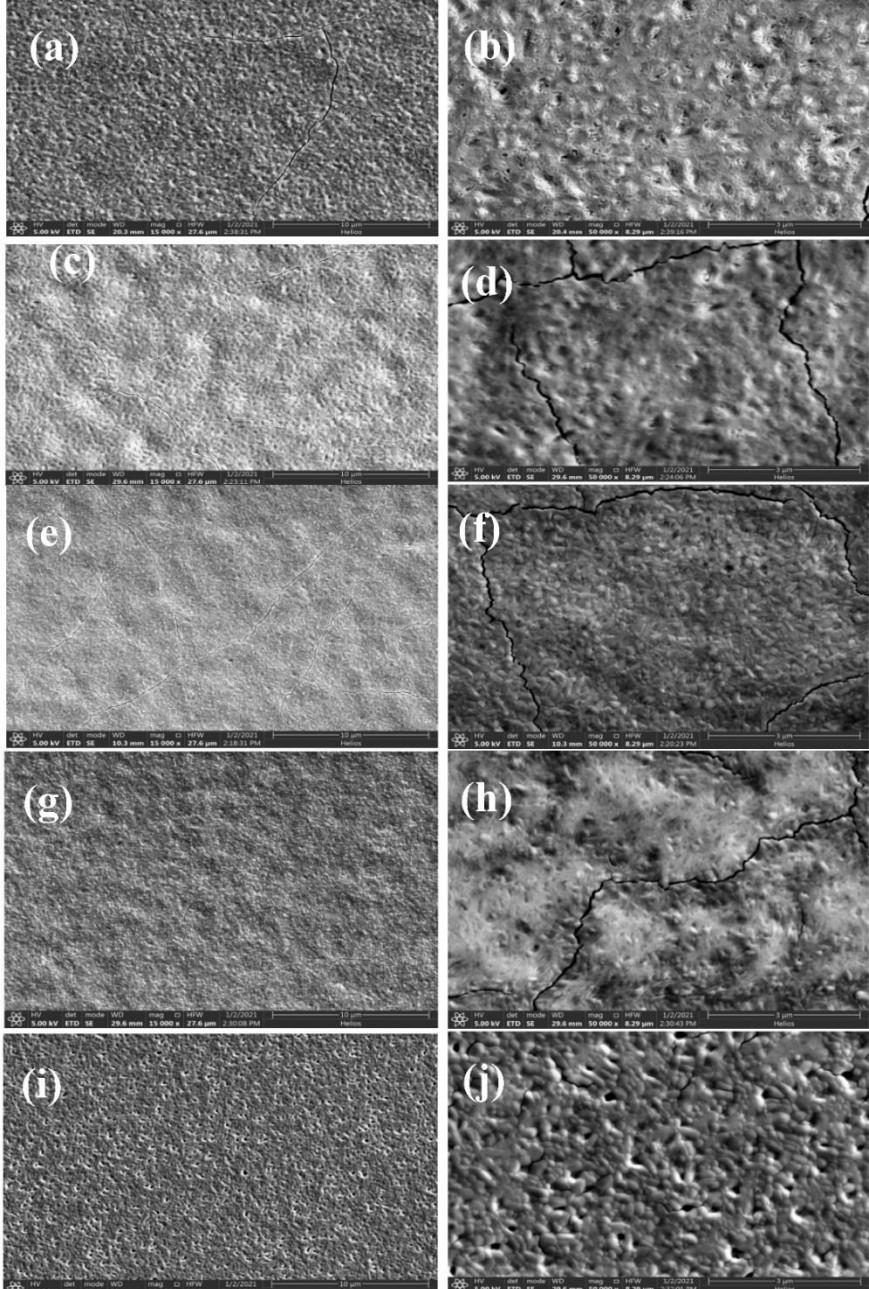

**Figure 3.** SEM top view of the multicomponent ferroelectric hybrid films. (**a**,**c**,**e**,**g**,**i**) are the ferroelectric films formed by $[C_6N_2H_{18}]BiI_5$ and $[C_6N_2H_{18}]SbI_5$ mixing ratios of 0.8:0.2, 0.6:0.4, 0.5:0.5, 0.4:0.6 and 0.2:0.8, in that order. (**b**,**d**,**f**,**h**,**j**) are enlarged views of (**a**,**c**,**e**,**g**,**i**), respectively.

From the analysis of the above morphology, it can be found that the film surface was dense and flat when the mixing ratio was 0.5:0.5, and the film quality was the highest. However, the presence of cracks may have degraded the performance of the devices prepared by this film. This may be related to the concentration of the solution, spin-coating time, spin-coating speed, and annealing conditions during the film preparation process, as the growth environment required for films with different mixing ratios varies. The morphology of the ferroelectric film with a mixing ratio of 0.2:0.8 was similar to that of the $[C_6N_2H_{18}]BiI_5$ film (Figure S1a), but the surface flatness was relatively poor and the growth of grains was irregular, leading to an increase in the number of holes.

### 3.4. Study of Photophysical Properties of Multicomponent Hybrid Ferroelectric Thin Films

Figure 4 shows the ultraviolet/visible (UV/Vis) absorption spectra of the multicomponent ferroelectric hybrid films. It can be seen from the figure that the absorption edges of these films were all in the range of 630 nm to 680 nm, which is the same trend as that of the two ferroelectric films of HDA-BiI$_5$ and [C$_6$N$_2$H$_{18}$]SbI$_5$ (Figure S2). In the range from 450 nm to 600 nm, the light absorption intensity of the ferroelectric films with a mixing ratio of 0.8:0.2 between Bi and Sb elements was slightly higher, and the absorption spectra of the other films in this range almost overlapped. This trend reflects the transition from [C$_6$N$_2$H$_{18}$]BiI$_5$ to [C$_6$N$_2$H$_{18}$]SbI$_5$ films, which can be well integrated with Figure S2. Because the absorbance values of pure HDA-BiI$_5$ and HDA-SbI$_5$ differed very little in the range of 400–550 nm, the absorbance values of the mixed materials almost overlapped after changing the concentration of the combination. It is noteworthy that, when the mixing ratio of Bi and Sb elements was 0.8:0.2, the position of the absorption edge of the film was significantly different from the other mixing ratios. With the increase in the atomic ratio of Sb elements, the absorption band edge of the film showed a tendency of red-shifting, even toward the long wavelength compared with the [C$_6$N$_2$H$_{18}$]BiI$_5$ film of Figure S2.

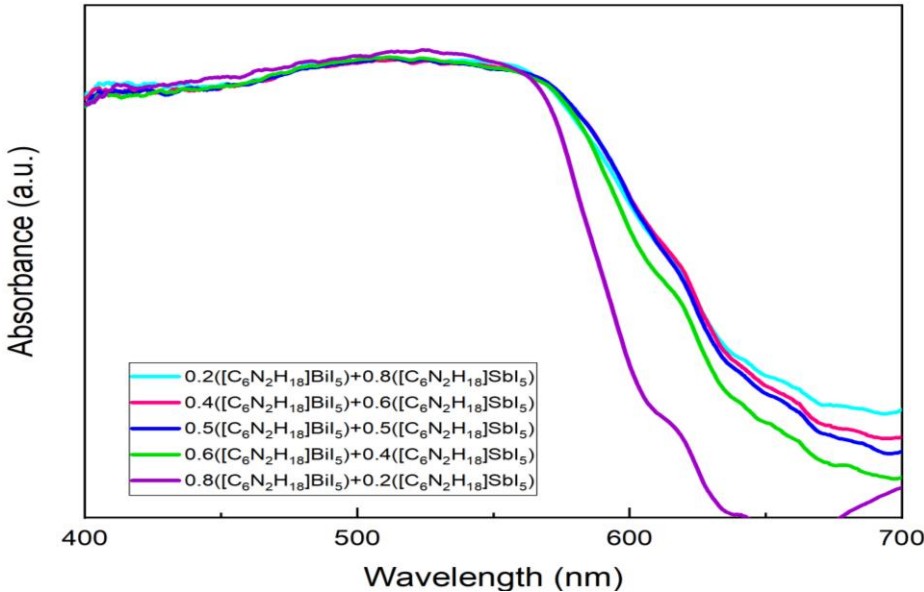

**Figure 4.** Ultraviolet/visible (UV/Vis) absorption spectra of multicomponent ferroelectric films.

In order to determine the bandgap of the multicomponent mixed ferroelectric films, the $(ah\upsilon)^{1/2} \sim h\upsilon$ relationship diagram is given in Figure 5, the calculation of which is described in the supporting literature and is not repeated here. From the figure, it can be observed that the bandgap of the ferroelectric film was about 2.02 eV when the mixing ratio of Bi and Sb elements was 0.8:0.2. The bandgap of the ferroelectric films gradually decreased with the increase in the atomic percentage of Sb elements, and the bandgaps of the films were 1.94 eV, 1.93 eV, and 1.91 eV when the mixing ratios of Bi and Sb elements were 0.6:0.4, 0.5:0.5, and 0.4:0.6, respectively. When the mixing ratio of Bi and Sb elements was 0.2:0.8, the bandgap of the ferroelectric film increased again to 1.95 eV. By comparison, it can be found that the mixing of two elements of group VA narrowed the bandgap of ferroelectric films (in most cases). A narrow bandgap means that the material can absorb a larger range of visible light, as shown in Figure 5, with an absorption band edge close to 700 nm, which has important implications for the study of optical absorption layers in ferroelectric photovoltaic devices. Optical absorption is closely related to the bandgap of the material. Solar power materials are generated by absorbing photons in electron–hole pairs so that electrons leap to the conduction band to become free electrons, generating a flow of electricity; hence, the incident light energy $h\upsilon$ must be greater than or equal to

the bandgap Eg of the material used. An appropriate reduction in the bandgap value can result in a higher utilization of light energy, which is conducive to light energy conversion.

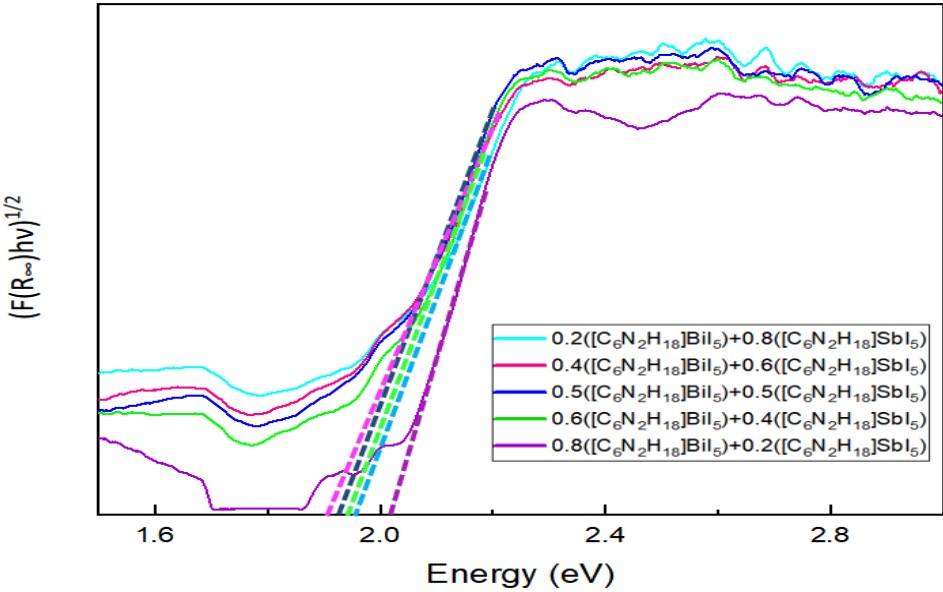

**Figure 5.** Plot of $(ah\upsilon)^{1/2} \sim h\upsilon$ for a multicomponent ferroelectric film.

We believe that the reason for bandgap narrowing is mainly due to the mixture of two ferroelectric materials, HDA-BiI$_5$ and [C$_6$N$_2$H$_{18}$]SbI$_5$, where Bi$^{2+}$ is replaced by Sb$^{2+}$, forming the [C$_6$N$_2$H$_{18}$]Bi$_{1-x}$Sb$_x$I$_5$ compound. Since the B-site divalent metal ions in the chalcogenide type material (ABX$_3$) occupy the main electronic orbitals at the bottom of the conduction band, it is obvious that B-site ion doping can tune the bandgap of chalcogenide materials. By adding Sb$^{3+}$ ions to increase the 6p electrons to fill the conduction band, the bottom shift of the conduction band was achieved, thus reducing the bandgap. It is noteworthy that the bandgaps of the films after mixing were narrower than those of the two original organic–inorganic hybrid films (Figure S3), a phenomenon that has also been observed in previous studies. One researcher reduced the bandgap of the films by doping 5% K ions into CH$_3$NH$_3$PbI$_3$, and this method reduced the bandgap of the films [10]. However, in 2014, Filip et al. calculated the bandgap of KPbI$_3$ as 1.70 eV using density flooding theory (DFT), which is higher than the bandgap of CH$_3$NH$_3$PbI$_3$ of 1.60 eV calculated by previous researchers using absorption spectroscopy [18].

*3.5. Calculation of Theoretical Bandgap Values for Multicomponent Hybrid Ferroelectric Materials*

The theoretical bandgap values of the hybrid ferroelectric materials were calculated using Materials Studio software, as shown in Figure 6. The specific calculation method and super cell structure diagram (Figure S4) are described in the supporting literature. As the two ferroelectric materials, HDA-BiI$_5$ and [C$_6$N$_2$H$_{18}$]SbI$_5$, were mixed, Bi$^{2+}$ was replaced by Sb$^{2+}$ to form the [C$_6$N$_2$H$_{18}$]Bi$_{1-x}$Sb$_x$I$_5$ compound. The crystal structure of this compound was modeled in MS software, the convergence calculations of the cutoff energy were performed, and the bandgap maps of the compound with different doping ratios were calculated and analyzed. As can be seen from Figure 6, the lowest point at the bottom of the conduction band and the highest point at the top of the valence band of the materials doped in various ratios were at the same position; hence, they were all direct bandgap semiconductors. The Fermi energy levels of each material were as follows: [C$_6$N$_2$H$_{18}$]SbI$_5$, 1.34 eV; 0.2 Bi, 2.03 eV; 0.4 Bi, 2.02 eV; 0.5 Bi, 1.71 eV; 0.6 Bi, 1.70 eV; 0.8 Bi, 2.01 eV; [C$_6$N$_2$H$_{18}$]BiI$_5$, 1.16 eV. When the ratio of Bi to Sb was 0.4:0.6, the narrowest bandgap of the compound was 1.58 eV. Compared with the experimental values of the bandgap in Figure 5, the bandgap values obtained from the theoretical calculations were smaller. This is due to the electronic autocorrelation problem of the pure DFT method, the absence of

derivative discontinuity in the exchange–correlation function, and the self-crossing error in the density generalized theory calculation, which usually lead to low prediction results for the bandgap of semiconductors or insulators. In addition, the actual preparation of hybrid films may lead to large experimental bandgap values because the quality of the final film is not completely ideal due to the environment and preparation techniques. However, the trend of the measured bandgap values in the actual experiments with the change in the ratio of Bi and Sb elements is consistent with the theoretical calculations. The purpose of calculating the theoretical bandgap is to obtain the trend of the bandgap as a reference basis for the experimental values, rather than specific values. It can be seen that the mixture of two ferroelectric materials, HDA-BiI$_5$ and [C$_6$N$_2$H$_{18}$]SbI$_5$, led to a decrease in the bandgap, which is more favorable for the absorption of visible light in the film.

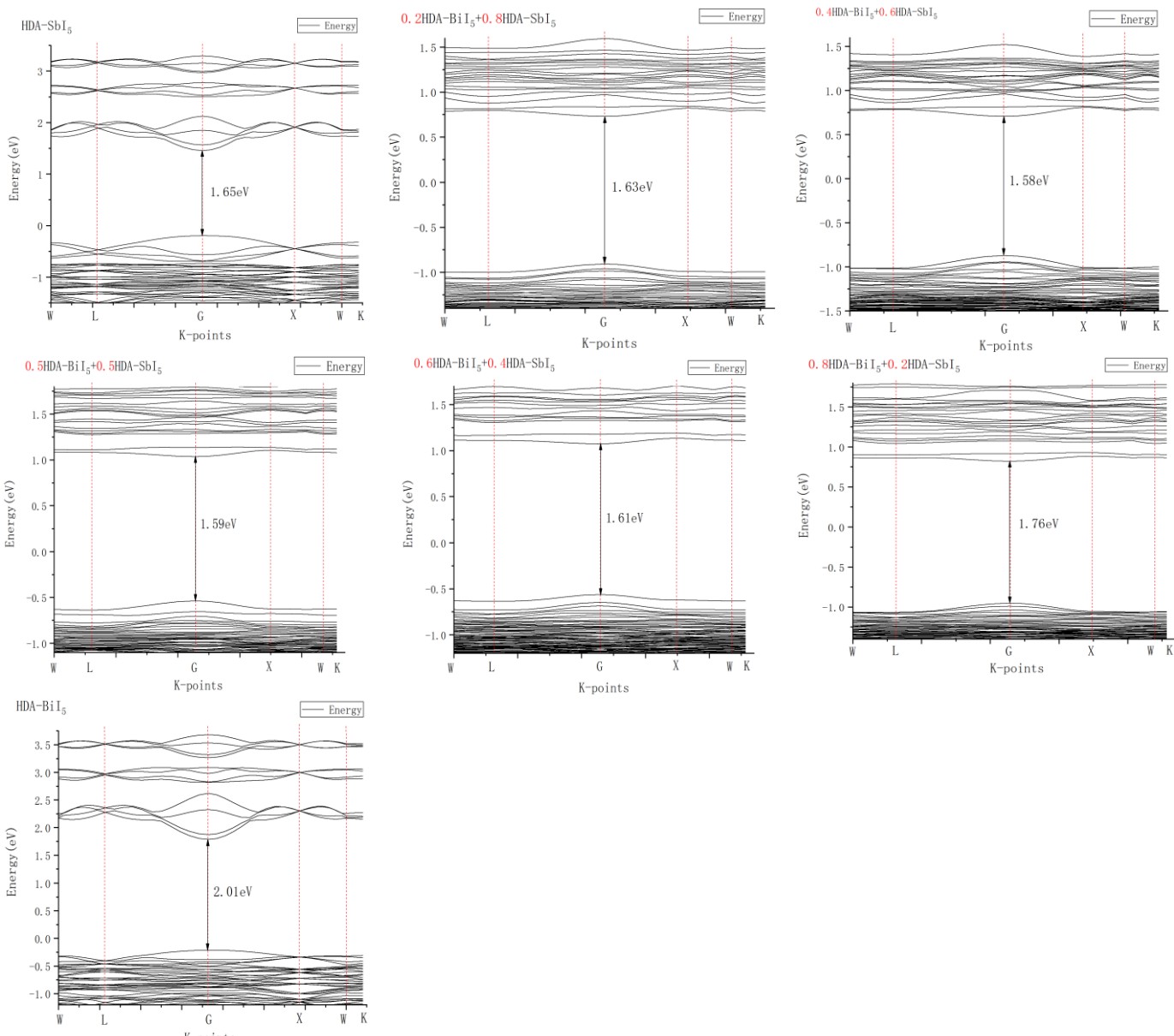

**Figure 6.** Theoretical calculated bandgap diagram of multicomponent ferroelectric films.

## 4. Conclusions

This paper investigated the preparation methods, crystal structures, surface morphology, and photophysical properties of different multicomponent organic–inorganic hybrid films. For the study of semiconductor properties, the bandgaps of different hybrid films

were calculated on the basis of the absorption edges of the absorption spectra. A theoretical model was developed to calculate the bandgaps of the mixed doping compounds with different concentrations for comparison with the experimental values. It could be calculated that the absorption edges of the hybrid films were red-shifted compared to the $[C_6N_2H_{18}]BiI_5$ and $[C_6N_2H_{18}]SbI_5$ films, indicating that the bandgap was reduced, consistent with theoretical values, which is more favorable for the absorption of visible light in the films and is important for the preparation of photovoltaic devices. When the mixing ratio of Bi and Sb elements was 0.5:0.5, the film surface was flat and dense, and the film quality was optimal. When the mixing ratio was 0.4:0.6, the film had the narrowest bandgap, which is most favorable for the absorption of visible light.

**Supplementary Materials:** The following supporting information can be downloaded at https://www.mdpi.com/article/10.3390/app122010454/s1: Figure S1. Top view of a single organic–inorganic hybrid ferroelectric film using scanning electron microscopy. (a,c) HDA-BiI$_5$ and $[C_6N_2H_{18}]SbI_5$ ferroelectric films, respectively. (b,d) Enlarged images of (a,c), respectively; Figure S2. UV/Vis absorption spectra and photoluminescence spectra of organic–inorganic hybrid ferroelectric thin films; Figure S3. *(ahv)$^{1/2}$~hv* relation of organic–inorganic hybrid ferroelectric thin films. (a) HDA-BiI$_5$; (b) $[C_6N_2H_{18}]SbI_5$; Figure S4. Optimized supercell diagram. (a) HAD-SbI$_5$. (b) 0.2Bi + 0.8Sb. (c) 0.4Bi + 0.6Sb. (d) 0.5Bi + 0.5Sb. (e) 0.6Bi + 0.4Sb. (f) 0.8Bi + 0.2Sb. (g) HAD-BiI$_5$.

**Author Contributions:** Conceptualization, J.Z. and X.Z.; methodology, Z.Z. and J.W.; software, W.H., H.Z. and H.W.; validation, J.Z., X.Z. and X.G.; formal analysis, X.G.; investigation, X.G.; resources, X.G. and H.W.; data curation, X.G.; writing—original draft preparation, X.G.; writing—review and editing, J.Z.; visualization, X.G. and H.Z.; supervision, J.Z.; project administration, C.Z. All authors have read and agreed to the published version of the manuscript.

**Funding:** This work was supported by the National Natural Science Foundation of China (No. 61875186 and No. 61901009) and State Key Laboratory of Advanced Optical Communication Systems Networks of China (2021GZKF002).

**Institutional Review Board Statement:** Not applicable.

**Informed Consent Statement:** Not applicable.

**Data Availability Statement:** The data used to support the findings of this study are available from the corresponding author upon request.

**Acknowledgments:** The authors thank Junming Li, Jin Cheng, Xingyao Chen, Shixian Huang, Mingkai Gu, and Weimin Wang for their help with this article.

**Conflicts of Interest:** The authors declare no conflict of interest.

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
