# Peer review of "Effect of Doping on the Bandgap of the Organic–Inorganic Hybrid Ferroelectric Material [C6N2H18]Bi1−xSbxI5 (0.0 < x < 1.0)"

_applsci, doi:10.3390/app122010454_

Round 1
Reviewer 1 Report
The authors report the synthesis and characterization of some lead-free organic halide perovskite thin films for solar cell applications, combining experimental and theoretical (DFT) results. The scientific merit and presentation of the work is reasonable, and in my opinion the manuscript is worthy of publication following some improvements.
Firstly, the authors perform characterization of the atomic structures of their materials using X-ray diffraction, but they limit their analysis of the results to qualitative description of the Bragg peaks. Since the crystal structure of the materials is fairly simple, the authors should be able to perform Rietveld refinement in order to determine the atomic positions, or at least Le Bail refinement to determine the lattice constants. This will allow more in-depth structural analysis.
The DFT portion of the study also requires some improvement. Firstly, the authors are using DFT to determine the electronic band structure and band gap, but the "band gap problem" in DFT is well-known to prevent accurate determination of the gap without the addition of some beyond-DFT methods or empirical corrections (e.g. DFT+U). Therefore, it is not clear what quantitative results can be expected from the authors' approach. The authors could provide some more in-depth analysis of, for example, the character of the bands in order to yield some more value from their DFT calculations. I am also concerned that a k-point mesh of 1x1x1 is insufficiently dense to obtain a well-converged band structure. The authors should describe what kind of convergence tests were performed.
Reviewer 2 Report
The manuscript presents optical properties of [C6N2H18]BiI5 and [C6N2H18]SbI5 and their derivative thin films. The obtained properties were studied experimentally and theoretically. The manuscript seems okay but there are few issues which need to be addressed. I recommend this manuscript for a major revision. My comments are as follows.
1. The introduction of the manuscript needs to be improved. It does not justify and gives a sufficient background of the research.
2. The scale for the SEM images in Figure 3 is not clearly visible. Authors need to mark the scale in bigger format.
3. In line number 219, authors states that the calculated theoretical band gap values are smaller as compared to obtained experimental values due to non-ideal thin film quality. In reviewer’s opinion, it may be because of the absence of derivative discontinuity in the exchange correlation function and self-interaction error in the density functional theory calculations. That is the trend of the experimental and theoretical values are matching. Authors should consider mentioning it in the manuscript.
4. In Figure 6, authors should consider explaining where the Fermi energy level in the band structure is. If it is at 0 eV, then mention it on the figure.
5. The manuscript has shown that the band gap value decreases by a significant margin for the 0.2HDA-BiI5+0.8HDA-SbI5 (from 2.01 eV to 1.63 eV). Authors should explain the reason for other compositions also in the manuscript.
6. Authors need to prepare a table comparing the band gap values obtained from theoretical and experimental methods with literature.
7. Why there is no significant change (from 400 to 550 nm) in the absorbance values after varying the concentrations of the compositions? Authors need to explain this also.
8. For Fig. 6, authors need to explain the calculated band gap is direct or indirect band gap. It should be explained in the manuscript.
9. For the theoretical part, authors should consider showing the figure for optimized supercell in the supplementary file.
Round 2
Reviewer 2 Report
This manuscript has been modified in accordance with the suggestions. The manuscript is recommended for publication in its present form.